# Dynamic displacement measurement of a wind turbine tower using accelerometers: tilt error compensation and validation

Clemens Jonscher[1], Paula Helming[2], David Märtins[1], Andreas Fischer[2], David Bonilla[1], Benedikt Hofmeister[1], Tanja Grießmann[1], and Raimund Rolfes[1]

[1]Leibniz University Hannover / ForWind, Institute of Structural Analysis, Appelstraße 9A, 30167 Hannover, Germany

[2]University of Bremen, Bremen Institute for Metrology, Automation and Quality Science, Linzer Str. 13, 28359 Bremen, Germany

**Correspondence:** Clemens Jonscher (c.jonscher@isd.uni-hannover.de)

**Abstract.** For vibration-based structural health monitoring (SHM) of wind turbine support structures, accelerometers are often used. Besides the structural acceleration, the measured quantity also contains the acceleration component due to gravity, which is known as tilt error. This tilt error must be quantified and taken into account, otherwise it can lead to incorrect evaluations, especially in the fatigue estimation or the dynamic displacement estimation using accelerometers. The standard solution is to explicitly measure the tilt angle, which requires an additional sensor for each measurement point and is not applicable for already recorded measurements without tilt information. Therefore, a novel tilt error compensation method is presented by using the static bending line. As a result the influence of the tilt error can be estimated in advance and no additional sensors for tilt measurement are needed. The compensation method is applied to accelerometer measurements of an onshore wind turbine tower and validated with contactless absolute distance measurements from a terrestrial laser scanning (TLS) system. The position and frequency-dependent tilt error of the investigated tower has a significant influence on the quasi static motion below 0.2 Hz with a minimum amplitude error of 9 %, whereas the normalised bending mode shapes around 0.3 Hz are only slightly affected.

## 1 Introduction

The towers of wind turbines are exposed to fluctuating forces caused by the rotor motion and wind loads. For remaining useful life (RUL) estimation, the real loads occurring on the towers are of interest. Currently, this evaluation is mostly done using strain gauges determining the local loads. In order to obtain more global information about the loads on the structure based on fewer sensors, methods such as modal expansion (Maes et al., 2016) can be used. Since strain gauges are considered susceptible to failures in long term measurements (Maes et al., 2016), displacement sensors can be used instead. Various approaches for the direct measuring of tower displacements exist, such as global navigation satellite system (GNSS) (Botz, 2022), laser doppler

vibrometry (LDV) (Dilek et al., 2019; Zieger et al., 2020), photogrammetry (Baqersad et al., 2017; Ozbek et al., 2013), motion tracking using a video camera (Helming et al., 2021), and terrestrial laser scanning (TLS) (Schill and Eichhorn, 2016; Artese and Nico, 2020; Helming et al., 2021). Except for GNSS, which in the case of Botz (2022) indicates unrealistically high quasi-static deformations for the top of the tower without specifying the quality flag or standard deviation, all other approaches require a reference point. This is a particular difficulty for offshore applications. The optical measurement method LDV also has the disadvantage that it requires reflective markings on the measurement object and only measures in the direction of the laser (Dilek et al., 2019; Zieger et al., 2020), while the photogrammetry (Baqersad et al., 2017; Ozbek et al., 2013) needs resolveable image features on the object of interest. In contrast, TLS needs no visible image features and no preparation of the measurement object. TLS has been shown to be capable of measuring tower deformation by positioning the laser scanner vertically (Schill and Eichhorn, 2016; Artese and Nico, 2020) or horizontally (Helming et al., 2021), while the latter includes a comparison with displacement measurements by means of motion tracking using a video camera.

Another possibility to determine displacements involves accelerometers attached to the monitored structures. Accelerometers are usually used for the vibration based structural health monitoring (SHM) of large structures to determine modal parameters such as natural frequencies and mode shapes (Devriendt et al., 2014; Häckell and Rolfes, 2013; Jonscher et al., 2024; Oliveira et al., 2018). There are other monitoring approaches using different sensor technology as described by Civera and Surace (2022). However, the VDI 3834 guideline recommends using the nacelle acceleration to estimate the health status using vibration measurements. Accelerometers are therefore already installed in many wind turbines. Since accelerometers are considered very robust, an estimation of the displacement by using acceleration measurement data has been a topic of research for a long time (Berg and Housner, 1961). For example, Camargo et al. (2019) applied time integration to measured data of a prototype concrete wind turbine tower. Park et al. (2005) estimated the displacement of a bridge using accelerometers. According to Stiros (2008), the measurement time, sampling rate and noise level of the measurement technology have the greatest influence when estimating displacement from acceleration measurement data. In the context of wind turbine towers, there are further recent studies (Jonscher et al., 2022b; Botz, 2022; Jonscher et al., 2022a) that investigate methods to obtain displacements from acceleration measurements. Jonscher et al. (2022b) showed that estimated dynamic displacements with calibrated Integrated Electronics Piezo Electric (IEPE) accelerometers and Micro-Electro-Mechanical-Systems (MEMS) accelerometers in the frequency range down to $0.05\,\mathrm{Hz}$ yield similar results. For both sensor types, high unexpected dynamic displacements in the very low frequency range were noticeable. Botz (2022) compared displacements from accelerometers with optical sensors in the frequency range from $0.2\,\mathrm{Hz}$ to $10\,\mathrm{Hz}$. The displacements obtained using accelerometers were systematically higher than those obtained by the optical sensors. The most likely cause of overestimation of dynamic displacements from acceleration measurements is the tilt error (Jonscher et al., 2022a).

The tilt error is caused by a temporal change in the alignment of the measurement direction relative to the Earth's gravity field, so that a part of the acceleration due to gravity is added to the structural acceleration measured by the sensor (Tarpø et al., 2021), leading to an overestimation. This effect can also be utilised to determine inclination with a tri-axial DC MEMS accelerometer (Łuczak, 2014). In the quasi-static case a tower deflects according to the static bending line, so that translatory motion at the tower top is always coupled with a slight tilt of the structure, resulting in a change in orientation relative to the Earth's gravity field. Basically, the tilt error can be corrected using the tilt angle (Boroschek and Legrand, 2006) before integrating the signal in time, such as determining the tilt angle using a tri-axial MEMS sensor. However, the direct measurement of the tilt is not always possible, as vibrations can falsify it. Therefore, Tarpø et al. (2021) developed a geometrical approach to compensate for the tilt error in a laboratory setting. However, if only one bi-axial accelerometers per measuring level is provided in the measurement setup, regardless of the sensor type, a virtual sensing concept is required. Since the tilt error is position-dependent on a flexible structure, it can lead to incorrect an identification of mode shapes and overestimation of low-frequency vibration amplitudes. As a consequence of the overestimation of displacement, the structural loads are also overestimated. Although the tilt error occurs in any structure where the orientation of the sensors to the Earth's gravity field changes in the low-frequency range, the tilt error plays a particular role in wind turbines in contrast to other tower structures. This fact is caused by the high head mass, the large area exposed to wind in relation to the cross-section of the tower, as well as operational changes to the wind turbine due to, e.g. the yaw movement. These excitations are particularly low-frequency, leading to a significant tilt error. Therefore, Tarpø et al. (2022) used a virtual sensing approach to estimate the dynamic strain of an offshore wind turbine tower using geophones and removed the tilt error by estimating the tilt angle using tri-axial geophones. So far, the influence of the tilt error on acceleration measurements of wind turbine towers is usually not considered. Furthermore, a predictive compensation method that functions without direct tilt angle measurement and the installation of respective additional sensors yet to be proposed. Such a method is especially relevant for existing measurement set ups, like the first German offshore wind farm Alpha Ventus.

In this study, a novel tilt error compensation method is introduced for tower structures based on the static bending line. This method allows a preliminary estimation of the influence of the tilt error and is tested in a field trial at the top of an onshore wind turbine tower providing dynamic displacement measurements using calibrated IEPE accelerometers down to 0.01 Hz. For the validation of the method, the displacement results are compared with TLS-based measurements. The accelerometer technology, time integration, novel tilt error compensation approach based on the static bending line as well as the TLS measurement principle are described in Section 2. The experimental setup and the wind turbine are presented in Section 3. Section 4 explains the mechanical model for determining the bending line and the theoretical influence on the measurements is examined. Subsequently, the application of the tilt error compensation to a wind turbine tower is presented in Section 5. In addition, the validation of the tilt error compensation by means of a TLS is carried out. In the

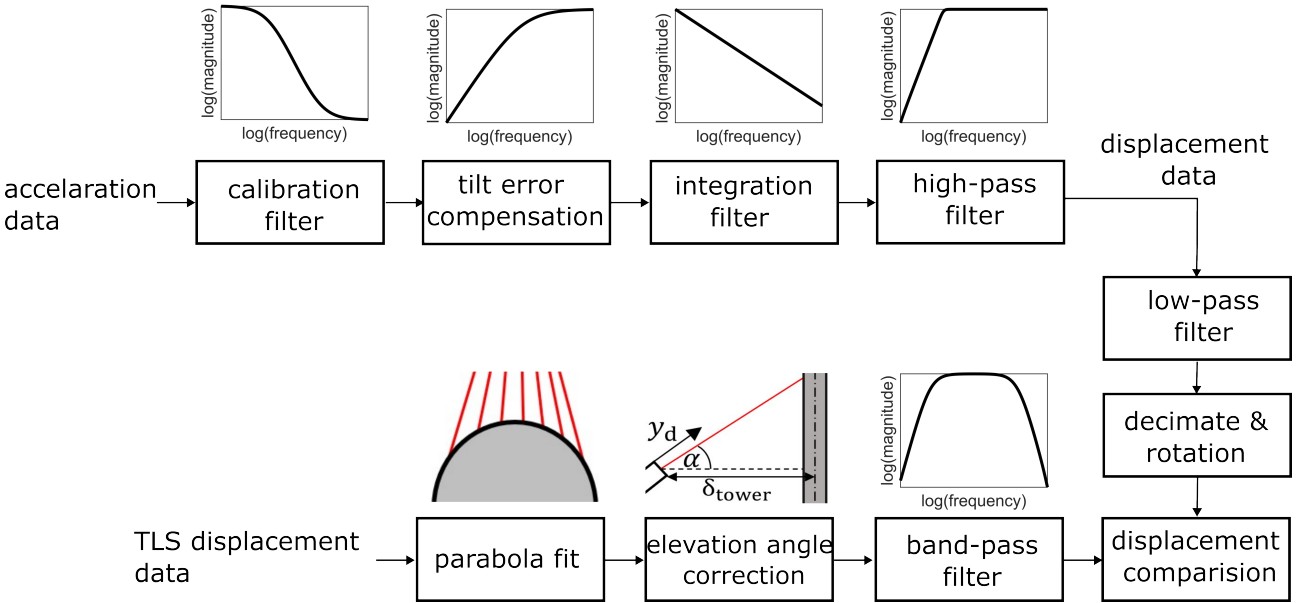

**Figure 1.** Flowchart of the dynamic displacement determination from acceleration measurement data taking into account the tilt error compensation based on the static bending line and the validation measurement with the TLS.

following Section 6 the benefits and limitations of the study are presented. Finally, Section 7 summarises the results and gives an outlook.

## 2 Theory

This study describes how low-frequency displacements of towers can be estimated using accelerometers, taking into account the tilt error. The validation is done using a TLS measurement. The corresponding flowchart of the process is shown in Fig. 1. The required theory of the various process steps is described in terms of accelerometers, integration filters and the compensation of the tilt error based on the static bending line. The functionality of the TLS is also briefly explained.

## 2.1 Accelerometers

Accelerometers are inertial sensors. This means that they do not require a reference measuring point. Accelerometers can be classified according to whether they can measure constant acceleration or not. Sensors that are sensitive to constant acceleration are also called DC sensors. MEMS capacitive accelerometers are widely used for this purpose. In contrast, AC sensors cannot measure constant acceleration. Common repre-

sentatives of this class are piezoelectric accelerometers. The measurement principle of the latter is based on the piezoelectric effect, which describes the coupling of electrical voltage and mechanical stress for special

crystals. Due to leakage currents, this type of sensor cannot measure constant acceleration. For better signal transmissibility, piezoelectric sensors with integrated electronics, also known as IEPE accelerometers, are often used. The amplifier integrated in the IEPE accelerometer is powered by a supply, which often has a decoupling high-pass filter. This is another reason that constant acceleration cannot be measured. The disadvantage of not being able to measure constant acceleration is offset by the advantage that piezoelectric sensors usually have a better noise floor above 1 Hz than comparable MEMS sensors. In order to be able to measure in the low-frequency range from 0.01 Hz to 1 Hz, a calibration is necessary for IEPE accelerometers, which can be performed using a centrifuge, as described in Jonscher et al. (2022b). In order to obtain a stable inverse filter for the calibration, the sensor is modeled as a shelved high-pass filter. To avoid a singularity at 0 Hz, a high-pass filter with a low cut-off frequency is used. This enables the signal behavior to be measured with phase and amplitude fidelity only down to a certain frequency.

### 2.1.1 Time integration of measured accelerations signals

For estimating the displacement from acceleration data, the data need to be integrated twice in time. The ideal transfer behaviour of a time integration filter is

$$H(j\omega) = \frac{1}{j\omega}. \tag{1}$$

Integration corresponds to a 90° phase shift and a frequency-dependent amplitude amplification of $\frac{1}{\omega}$, shown in Fig. 2 (yellow dotted line).

A double integration in time can be performed in the frequency domain by dividing the discrete Fourier transform by $-\omega^2$ (Brandt and Brincker, 2014). The spectrum can then be transformed back into the time domain using the inverse discrete Fourier transform. This method has an ideal transfer behaviour, but it is not real-time capable, because the signal needs to be transformed as a whole.

The time integration of the acceleration data can also be carried out in the time domain by using filters. The associated filters are causal filters and therefore real-time capable. The standard integration method is the Newton-Cotes formula of order one, also known as trapezoidal rule. It can be formulated as an infinite impulse response (IIR) filter

$$H(z) = \frac{\frac{1}{2f_s}z^{-1} + \frac{1}{2f_s}}{z^{-1} - 1}. \tag{2}$$

This method optimally matches the phase response of the ideal integration filter at the expense of amplitude accuracy, as shown in Fig. 2 (red line). This can be compensated by a sufficiently high sampling rate of the signal. The IIR integration filters are semistable, which leads to a drift in the time series. Therefore, the application of a high pass filter after the double time integration using the IIR filter is common practice in order to prevent drift and is performed such in this study. Due to missing initial conditions of integration and high pass filter,

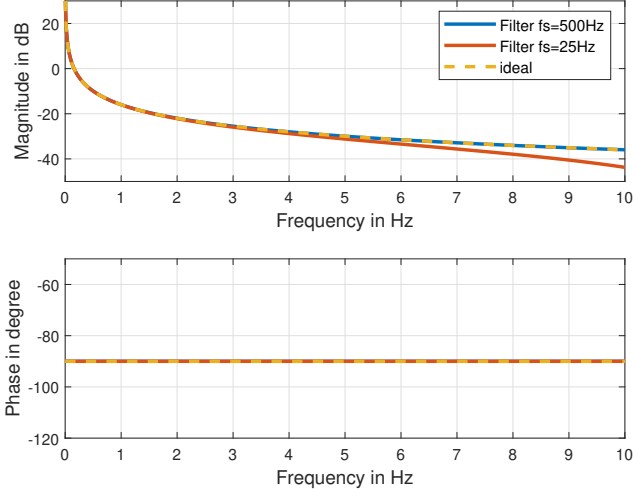

**Figure 2.** Transfer behaviour of the integration filters for different sampling frequencies $f_s$ compared to the ideal transfer behaviour.

only the calculation of a dynamic or relative displacement is possible. The lower the noise level of the sensors is, the lower is the frequency at which the displacement can be determined in a stable way.

### 2.1.2 Tilt error compensation

Accelerometers are inertial sensors, which means that the measured acceleration $\mathbf{a}_{meas}$ contains the gravitational acceleration $\mathbf{a}_g$ in addition to the structural acceleration $\mathbf{a}_{str}$:

$$\mathbf{a}_{meas} = \mathbf{a}_{str} + \mathbf{a}_g. \tag{3}$$

If the orientation of the sensors to the Earth's gravity field remains constant, it will not be picked up by AC accelerometers. If the angle $\mathbf{w}'$ of the sensors to the Earth's gravity field changes slightly, the proportion of the measured acceleration due to gravity changes with time, resulting in a so-called tilt error. For a one-dimensional horizontal sensor this results in

$$a_{meas} = a_{str} + \sin(w')g, \tag{4}$$

as shown in Fig. 3. The tilt error has a particularly strong effect in the very low frequency range, which is further significantly increased by time integration. Therefore, the tilt error must be compensated before time integration, otherwise very large errors will occur (Boroschek and Legrand, 2006). The simplest approach to compensate the tilt error is to measure the tilt angle $w'$, whereby care must be taken to ensure that the angle measurement is as insensitive as possible to acceleration. In the presented study, a novel virtual sensing approach for the estimation of the tilt angle is applied using the static bending line of the tower. This has the

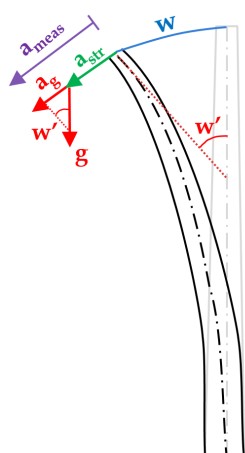

**Figure 3.** Sketch of the principle of the tilt error.

particular advantage that the influence of the tilt error can be estimated even before the actual measurement. The approach applies the small angle approximation $w' \ll 1$, which leads to

$$\sin(w') \approx w'. \tag{5}$$

For most tower structures, this is a reasonable simplification, since they do not exceed a few degrees of tilt under nominal operating conditions. In linear Bernoulli-Euler beam theory, the tilt angle is proportional to the displacement $w$

$$w' = wm, \tag{6}$$

where $m$ is the proportionality factor. The proportionality factor depends on the structure and is determined from static bending line under the assumption that the structure behaves like the static bending line in the low-frequency range. Usually, the factor is dependent on the position along the tower structure, so that for the determination, a model is required, which is described in more detail in Section 4. From the measured acceleration amplitude, one obtains the frequency-dependent displacement amplitude according by a double time integration, which according to Equation 1 corresponds to

$$w(f) = (2\pi f)^{-2} a_{\text{str}}(f). \tag{7}$$

Substituting into Equation 4 with the small angle approximation of Equation 5 and 6 leads to

$$a_{\text{meas}}(f) = a_{\text{str}}(f) + g\,(2\pi f)^{-2} a_{\text{str}}(f)m, \tag{8}$$

and the solution for $a_{\text{str}}$ yields

$$a_{\text{str}}(f) = a_{\text{meas}}(f)\frac{1}{(2\pi f)^{-2} gm + 1} = a_{\text{meas}}(f)c(f). \tag{9}$$

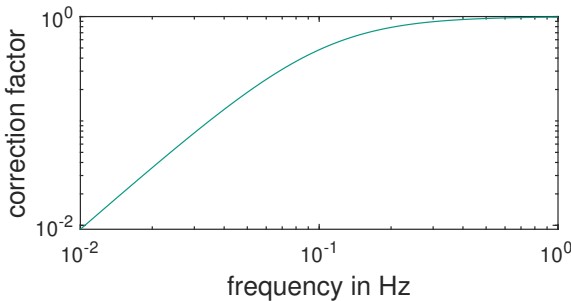

**Figure 4.** Double logarithmic representation of the frequency-depended correction factor of $a_{\mathrm{meas}}$ as a function of frequency for $gm = 0.4267 \; \frac{1}{s^2}$. There is no phase change due to the tilt error compensation.

As a result, a frequency-dependent correction factor $c(f)$ of the tilt-error compensation has to be multiplied by the measured amplitude to obtain the structural acceleration. The correction factor $c(f)$ depends on the acceleration due to the gravity $g$ and the proportionality factor $m$. The proportionality factor itself depends on the static bending line and is therefore dependent on the structure and measurement position. A qualitative behavior of the frequency-dependent correction factor is shown in Fig. 4, which corresponds to the amplitude response of a second-order high-pass filter with a cutoff frequency of $gm$, see Equation 9. It is important to note that unlike a normal high-pass filter, there is no phase shift due to the tilt error. These transfer characteristics can be achieved with a non-causal zero-phase filter. For this purpose, a causal first-order high-pass filter with the cutoff frequency of $gm$ is applied forward and backward in time (Gustafsson, 1996) on the measured acceleration signal $a_{\mathrm{str}}$.

## 2.2   Measurement Principle of Terrestrial Laser Scanning

To cross-validate the dynamic displacements determined by the accelerometers, the displacements are additionally measured using an alternative measurement approach. For this purpose, terrestrial laser-based distance measurements are performed on the tower of the wind turbine. The underlying measurement principle is time-of-flight, while the measurement method is also known as terrestrial laser scanning (TLS). In order to determine a distance $d$, a laser source emits a light pulse and the laser light travels to the measurement object. The laser light is scattered on the object's surface and a part of of the scattered laser light travels back in the direction to the TLS system, where a photo detector detects the incoming light pulse. With the known speed of light $c$ and the measured time $t$ from the laser pulse emission to the detection of the scattered light pulse, the distance $d$ follows from the equation

$$d = \frac{c \cdot t}{2}. \tag{10}$$

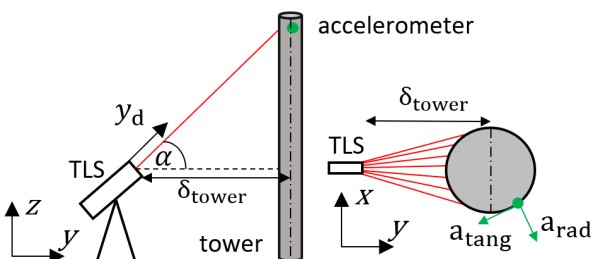

**Figure 5.** Sketch of the measurement setup. $\delta_{\text{tower}}$ is the distance from the TLS to the tower, $a_{\text{tang}}$ is the accelerometer measured tangential direction and $a_{\text{rad}}$ the radial direction.

To measure not only at a single position or one tower surface point, respectively, the laser beam is continuously moving, which results in a laser scanning. As a result, the distance to the measurement object is measured several times along the scan line. Here, the TLS is operated in the line-scanning mode, while the laser scans horizontally by means of a rotating mirror. The tower surface points are first obtained in polar coordinates and then transformed to two-dimensional Cartesian coordinates (axial/depth axis and lateral scan axis).

To extract the horizontal displacement of the tower in the axial direction $y$ (direction of measurement) and the lateral direction $x$ at the scanned tower height, the two-dimensional tower position is calculated using a least-squares fit of a parabolic function to the measured tower surface points, as described in Helming et al. (2021). Since the optical axis of the TLS system is not perpendicular to the tower axis, the elevation angle $\alpha$ according to the tilt of the TLS system with respect to the horizon needs to be taken into account. Therefore, the axial position $y$ of the observed tower section is derived from the initially measured position $y_d$ with the TLS system according to

$$y = y_d \cos(\alpha). \tag{11}$$

A visualisation of the measurement setup is shown in Fig. 5.

The inclination of the tower itself is assumed to be negligibly small. Further details about the TLS measurement approach including the system's capabilities and the achievable displacement measurement uncertainty to detect tower and rotor blade deformations are published in Helming et al. (2021) and Helming et al. (2023), respectively.

## 3   Experimental Setup

The investigation of the compensation of the tilt error of accelerometers based on the static bending line in comparison to displacement measurements using the TLS is carried out on the tower of an onshore wind

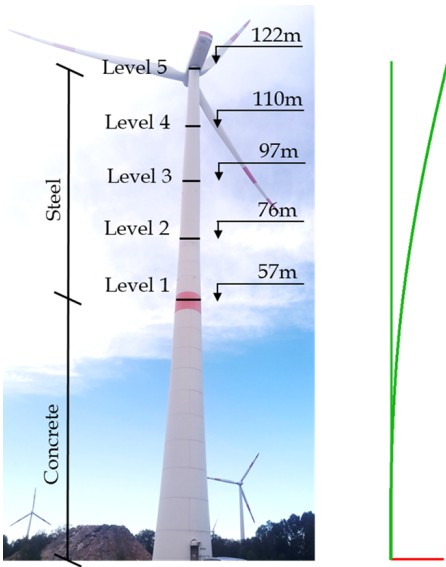

**Figure 6.** Left: Measurement levels of the investigated steel-concrete hybrid wind turbine tower, right: Finite element model of the tower in simulation tool DeSiO (design and simulation of offshore support structures) with and without static loading explained in Section 4.

turbine with a rated power of 3.4 MW, shown in Fig. 6. The 122 m tower is a hybrid contruction, with the first 57 m being prestressed concrete and the remaining 65 m being steel. The first natural bending frequency of the tower is around 0.3 Hz.

For vibration-based long term monitoring (Jonscher et al., 2024), IEPE accelerometers were installed to measure in two horizontal directions at each of the five measurement levels shown in Fig. 6. A 24 bit data acquisition device (DAQ) records all sensors with a sampling rate of 500 Hz synchronously. The relatively high sampling rate was chosen to avoid amplitude errors resulting from the integration filter (Section 2.1.1, Fig. 2). To enable the measurement of low-frequency signal components with IEPE accelerometers, a sensor supply with a high-pass cut-off frequency of 0.0106 Hz is used. The calibration was carried out using a centrifuge as described by Jonscher et al. (2022b). The resulting transfer behaviour and filter model of the IEPE accelerometers are shown as an example for the two sensors of the measurement level 5 in Fig. 7. Sensor 1 is attached radial to the tower and sensor 2 tangential. The corresponding filter settings are listed in Table 1. These calibrated accelerometers are used for comparing to the TLS.

For comparison, the displacement of the tower is measured approximately at the height of measurement level 5 using the TLS system. The evaluation is performed with a parabola fit on the scanned tower cross-section, which determines the displacement in the horizontal plane. A detailed description of the signal processing and the laser scanning system LASE 2000D-227 from the company LASE is contained in Helming et al. (2021). The

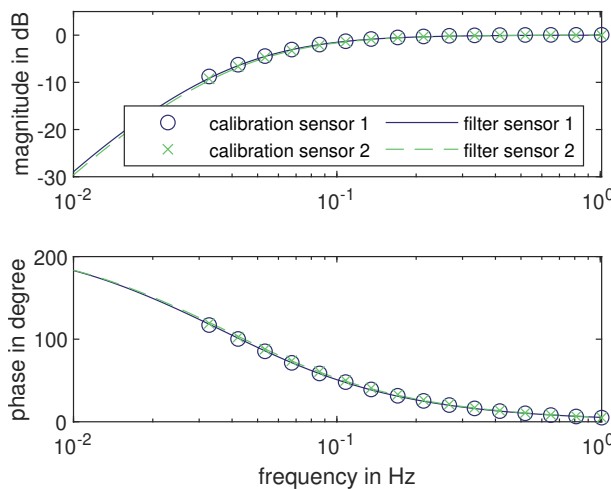

**Figure 7.** Calibration results from 0.033 Hz - 1 Hz with filter model of the radial (sensor 1) and tangential (sensor 2) accelerometers of measurement level 5.

**Table 1.** Calibration and high pass filter settings of the accelerometers determined according to Jonscher et al. (2022b).

|  | sensor 1 | sensor 2 |
| --- | --- | --- |
| $G_{\mathrm{shelf}}$ (dB) | -90 | -90 |
| $f_c$ sensor (Hz) | 2x0.0431 | 2x0.0450 |
| $f_c$ supply (Hz) | 0.0106 | 0.0106 |
| $G_{\mathrm{max}}$ (dB) | 30 | 30 |
| $f_c$ high pass (Hz) | 2x0.0057 | 2x0.0059 |

measurement setup of the laser is shown in Fig. 8. During the continuous acceleration measurements, the TLS is used at three separate time intervals to measure the dynamic displacement. The associated measurement, wind, and wind turbine parameters are listed in Table 2. Only this 10 minutes aggregated SCADA data are available during the measurement campaign.

## 4   Structural model of the tower to predict the tilt error

As described in Section 2.1.2, the static bending line of the tower is required to determine the proportionality factor $m$ to compensate for the tilt error of the acceleration measurement. For this purpose, the structural solver of the currently developed in-house simulation tool, DeSiO (design and simulation of offshore support structures) developed by Gebhardt et al. (2019), is used. The framework consists of the three canonical models rigid bodies, geometrically exact beams and solid degenerate shells, using a director-based total Lagrangian formulation. In addition, an energy conserving or controlled energy dissipating time integration method is

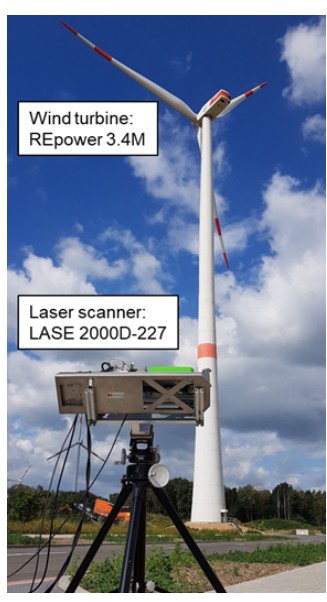

**Figure 8.** Setup of the in-field displacement measurement using a TLS.

**Table 2.** Investigated TLS displacement measurements for dynamic displacement comparison with accelerometers.

|  | Measurement 1 | Measurement 2 | Measurement 3 |
|---|---|---|---|
| Date | 2021-10-22 | 2022-10-10 | 2022-11-01 |
| Distance TLS to tower $\delta_{\text{tower}}$ [m] | 120 | 198 | 114 |
| Measurement direction TLS [°] | 261 W | 330 NW | 253 W |
| Inclination angle $\alpha$ [°] | 43.5 | 31.8 | 45.6 |
| Median laser points on tower | 8 | 4 | 6 |
| Averaged wind speed [$\frac{m}{s}$] | 12.53 | 9.3 | 11.2 |
| Averaged rotor speed [rpm] | 13.7 | 13.2 | 13.6 |
| Generated electrical power [kW] | 3373 | 2181 | 2690 |
| Nacelle position [°] | 242 SW | 249 SW | 228 SW |

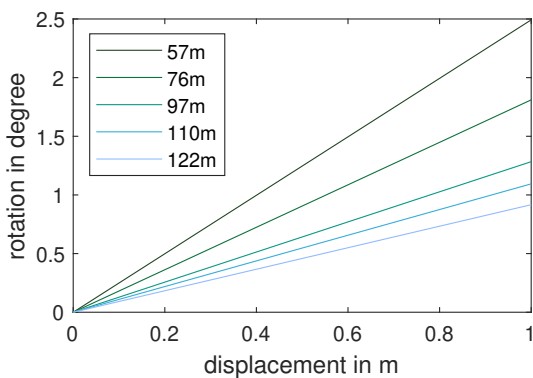

**Figure 9.** Correlation between displacement and rotation at the measurement levels on the wind turbine tower.

**Table 3.** Proportionality factors $m$ for each measurement levels

| measurement level | height [m] | $m\ [\frac{1}{m}]$ |
|:---:|:---:|:---:|
| 1 | 57 | 0.0435 |
| 2 | 76 | 0.0316 |
| 3 | 97 | 0.0224 |
| 4 | 110 | 0.0191 |
| 5 | 122 | 0.0160 |

implemented, which prevents the accumulation of errors due to numerical integration in long simulations
(Gebhardt et al., 2020).

Slender and flexible components of a wind turbine, such as the tower or the blades, are idealised by geometrically exact beams. Elements that are not expected to deform significantly, such as the nacelle and hub, are simplified using rigid bodies. Since, in the present case, only the static bending of the tower is of interest, the blades, hub and nacelle are considered to be a point mass (222.80 $t$) on the tower top. Soil stiffness is neglected
and the cross-section properties were determined from the tower geometry and materials.

The tower is discretised using 43 elements. It is ensured that there is a node at each of the measuring points and thus the displacements and rotations can be determined without interpolation. Static calculations with different constant loads at the top of the tower were carried out. The loads were distributed in the range of $5.0 \cdot 10^5\ N$ to $2.0 \cdot 10^7\ N$ and applied in the horizontal direction. In Fig. 6, a visualisation of the finite element model is shown,
both before and after static loading. The resulting relationship between displacement and rotation for each measurement level is shown in Fig. 9. The linear relationship allows a straight line equation with a slope $m$ to be determined for each measurement level. The slope is equal to the proportionality factor $m$ introduced in Section 2.1.2 and the results are listed in Table 3.

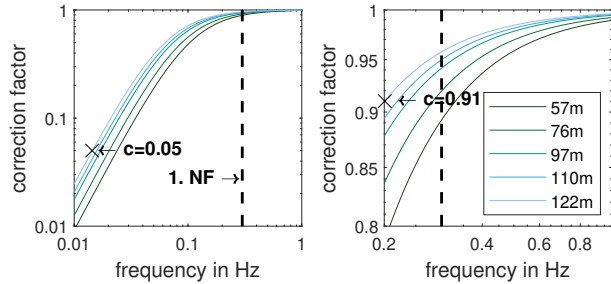

**Figure 10.** Frequency-dependent tilt error correction factors for the different measurement levels based on the DeSiO model, dashed line showing the first bending natural frequency (NF). left: Frequency range between 0.01 Hz and 1 Hz right: Frequency range close to the first natural frequency between 0.2 Hz and 1 Hz.

With the proportionality factor values, the resulting frequency-dependent correction factors can be determined by means of Equation 9. These correction factors are shown in Fig. 10. Basically, this analysis shows that for methods that require an exact acceleration amplitude, like the dynamic strain estimation using acceleration measurements, the tilt error should be taken into account below 1 Hz for this particular tower. It is essential for signal components of less than 0.2 Hz with a maximum correction factor of 0.91 corresponding to a minimum amplitude error of 9%. For very low frequencies, Fig. 10 demonstrates, that the correction factor of the highest measuring level drops below 0.05. This means that 95% of the measured acceleration amplitude is due to tilt and only 5% are due to the actual structural acceleration. The impact of these errors can be exemplified for tower fatigue life estimation as a typical application for structural measurements. The fatigue life ratio between the corrected and raw measurement data can be expressed, assuming small deformations, a linear stress-strain relationship and a harmonic acceleration signal, using an equation inspired by the Eurocode 3 (EN 1993-1-9:2005).

$$
\frac{N_{\mathrm{corr}}}{N_{\mathrm{raw}}} = \left( \frac{\Delta\sigma_{\mathrm{raw}}}{\Delta\sigma_{\mathrm{corr}}} \right)^m = \left( \frac{a_{\mathrm{raw}}}{a_{\mathrm{corr}}} \right)^m = \left( c(f) \right)^m , \tag{12}
$$

where $m = 3$ for steel structures and $\frac{a_{\mathrm{raw}}}{a_{\mathrm{corr}}}$ corresponds to the correction factor $c(f)$. Under the assumptions outlined above, the ratio between the measured and corrected stresses $\frac{\Delta\sigma_{\mathrm{raw}}}{\Delta\sigma_{\mathrm{corr}}}$ also corresponds to $c(f)$. Using the correction factor $c(f) = 0.91$ results in $\frac{N_{\mathrm{corr}}}{N_{\mathrm{raw}}} = 0.75$, which means that using the uncorrected data results in 25% reduced fatigue life estimates. Considering $c(f) = 0.05$ results in $\frac{N_{\mathrm{corr}}}{N_{\mathrm{raw}}} = 0.00013$, which shows that for very low frequencies, the raw data is essentially unusable for fatigue life estimations.

For evaluations that require relative quantities, such as the normalised first bending mode shape with a natural frequency of 0.3 Hz, there is also an influence, but it will be small. The counterintuitive result is that lower measurement levels are reduced more due to a higher proportionality factor $m$ and therefore appear to have a greater tilt error. However, these are relative magnitudes. As the larger vibration amplitudes are to be expected at the top of the tower, the absolute tilt error is highest there.

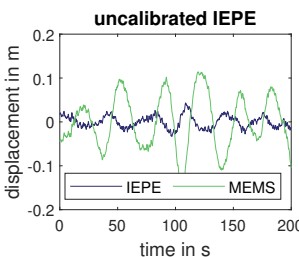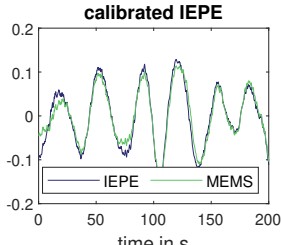

**Figure 11.** Sensor comparison between MEMS and IEPE accelerometers for displacement estimation using time integration.

## 5 Application of the tilt error compensation to acceleration measurements of a wind turbine tower

This section presents the application of calibrated IEPE accelerometers for the determination of dynamic displacements considering tilt compensation to a wind turbine tower. Finally, a validation of the dynamic displacement measurements from accelerometers with TLS displacement measurements is performed.

### 5.1 Quantification of the tilt error influence

Jonscher et al. (2022a) carried out a sensor comparison between DC-capable MEMS and IEPE accelerometers in a preliminary experiment close to measurement level 3 at a height of 96 m. The corresponding result of the comparison is shown in Fig. 11. It exemplifies that after calibration and double time integration of the acceleration signals, both sensor types show the identical high vibration amplitudes in the low frequency range. Thus, the effect of the tilt error is demonstrated to be independent of the sensor type.

The tilt error compensation using the DeSiO model described in Section 2.1.2 provides position-dependent correction factors, which are shown in Fig. 10. The tilt error has an influence in the range of the first bending natural frequency. As a result, without considering the tilt error, the vibrations of the first bending mode pair are measured with a position-dependent error. The effect on the normalised mode shape $\phi$ shown in Fig. 12 is negligible, since the normalisation removes most of the deviations introduced by the tilt error. However, a small overestimation below 1% of the normalised mode shape at the upper level and an underestimation at the lower level below 5% can be observed. For structures with more curvature, the influence of the tilt error on the mode shape can be significantly greater, as Tarpø et al. (2021) have shown for a laboratory structure. The tilt error has a significantly higher influence on vibrations below the natural frequency in the examined wind turbine tower. To demonstrate this, the displacements from the accelerometers in the frequency range 0.01-2 Hz are determined with and without tilt error compensation in the measurement direction of the lasers during the period of the first TLS measurement. The bandpass filter is used to ensure that only frequency components relevant for the evaluation are included in the time signal by reducing measurement noise. The results are shown in Fig. 13. The displacements are significantly larger without the tilt error compensation. A

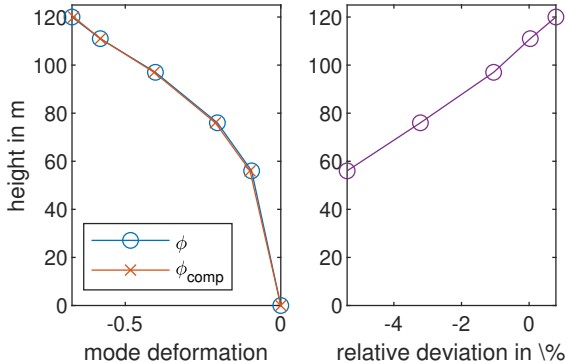

**Figure 12.** Influence of the tilt error on a normalised mode shape $\phi_{\mathrm{comp}}$ and without $\phi$ the compensation identified using Bayesian operational modal analysis (BAYOMA) (Jonscher et al., 2024), left: mode shapes, right: relative deviation $(\phi - \phi_{\mathrm{comp}})/\phi$ between both mode shapes.

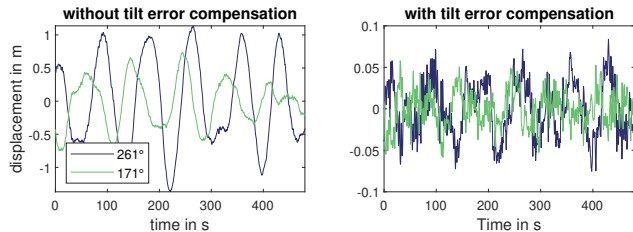

**Figure 13.** Estimated dynamic displacements from accelerometers in the frequency range 0.01-2 Hz with and without tilt error compensation. Note the different scaling of the displacement axis.

very low frequency component near 0.01 Hz dominates in both directions. This can be attributed to fluctuations in the wind speed and operating condition of the turbine. With tilt error compensation, the amplitude of the dominant frequency is significantly reduced by about 95% and other vibration components become visible. To verify this high reduction, a validation by direct displacement measurement is necessary, which is described in the following section.

### 5.2 Validation with optical displacement measurements

The validation of the tilt error compensation presented in Section 2.1.2 on the basis of the first bending line is carried out using a TLS. For direct comparison, the signals are low pass filtered and decimated to a sampling rate of 5 Hz after the displacement calculation. The coordinate system of the dynamic displacements from the accelerometers is then rotated in the direction of the laser measurement. The time curves of the comparison of the TLS without a high pass and the accelerometers in a frequency range 0.005-2 Hz are shown in Fig. 14. The basic characteristics of the signal seem to be correct. However, there are low-frequency oscillations in

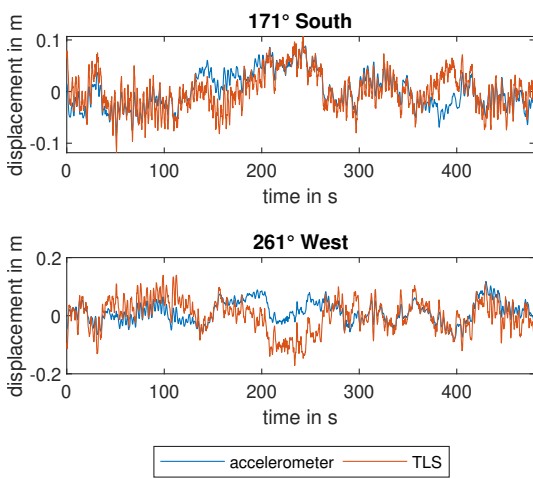

**Figure 14.** Comparison of the displacements determined from accelerometers (frequency range 0.005-2 Hz) and TLS (frequency range 0-2 Hz) for TLS measurement 1 with a averaged wind speed of 12.53 m/s and power of 3373 kW.

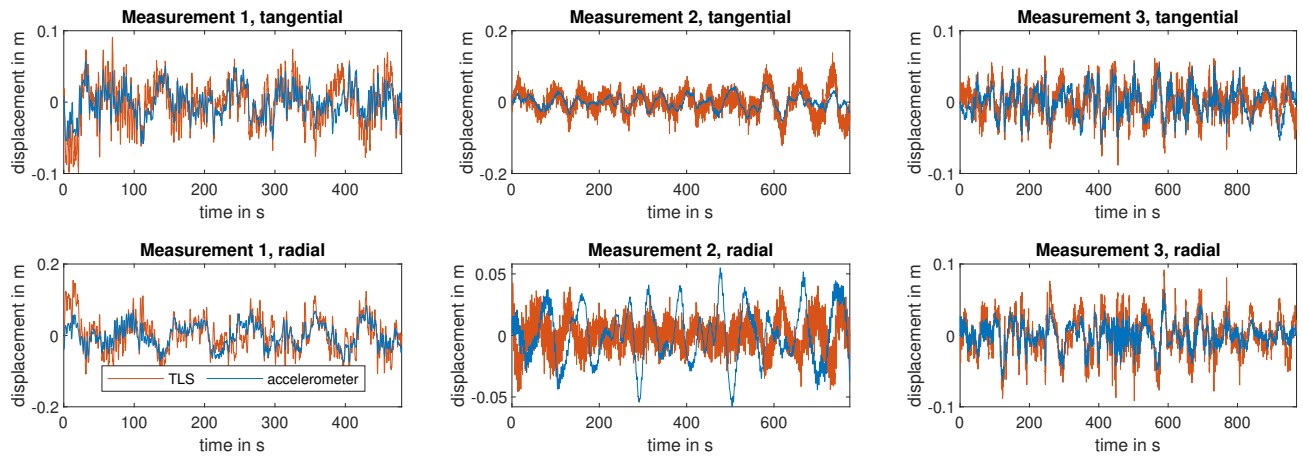

**Figure 15.** Comparison of the displacements determined with accelerometers and TLS for all measurements in the frequency range from 0.01-2 Hz.

the signal of the laser that are not correctly mapped by the accelerometers. The reasons, apart from the high pass filter of the accelerometers, are that the accelerometers are only calibrated down to 0.033 Hz and the calibration filter model only reliably provides correct amplitudes and phases above this frequency.

A comparison of the displacement signals from the accelerometers and the TLS with a zero phase high pass filter with a cut off frequency of 0.01 Hz is shown for all measurements in Fig. 15. With the exception of measurement 2 in the radial direction of the laser, the results between the laser and the displacements appear very similar. A comparison based on the Pearson correlation coefficient and the normalised mean square error

**Table 4.** Correlation coefficient $\rho$ and the normalised mean square error (nMSE) of the displacement given by TLS and acceleration measurements with and without tilt error compensation.

| | | Measurement 1 | | Measurement 2 | | Measurement 3 | |
|---|---|---|---|---|---|---|---|
| | | tangential | radial | tangential | radial | tangential | radial |
| with compensation | $\rho$ | 0.7808 | 0.8040 | 0.7299 | 0.0057 | 0.7489 | 0.8862 |
| | $nMSE$ | 3.5546 | 8.4277 | 0.1172 | 14.2285 | 0.6877 | 0.3109 |
| without compensation | $\rho$ | 0.4428 | 0.5464 | 0.6290 | -0.0846 | 0.3104 | 0.6267 |
| | $nMSE$ | 744.8 | 251.6 | 71.8 | 6318.1 | 601.5 | 28.2 |

(nMSE), given by:

$$\mathrm{nMSE} = \frac{1}{N\sigma^2_{w_{laser}}} = \sum_{i=1}^{N}\left(w_{laser,i} - w_{accel,i}\right)^2, \tag{13}$$

is listed for all three measurements in Table 4 for the case with and without the presented tilt error compensation of the acceleration measurement. A Pearson correlation coefficient closer to 1 indicates that the signals are correlated, with a value close to 0 the signals are uncorrelated. Tilt error compensation makes the signals

- with the exception of measurement two in radial direction - significant more correlated. However, this metric cannot provide any information about an amplitude error. The nMSE is used for this purpose. It is clearly evident that the error is greatly reduced by the proposed tilt error compensation method. Therefore, it can be concluded that the presented tilt error compensation is valid. A cause for the error in the second radial measurement has not yet been found. Here, the accelerometer displacement has a much more dominant low

frequency signal component. Basically, it seems that the laser has more high frequency signal components than the accelerometers.

A more precise indication of this is given by the auto power spectral density of all measurements shown in Fig. 16. It can be seen that the noise level of the laser above 1 Hz is significantly higher than that of the accelerometers. In addition, the tangential measurement direction of the laser is noisier than the radial

direction, and the fewer measurement points the laser has on the tower, the higher the noise level. Basically, the laser measurements seem to have a slightly higher signal power, except for measurement 2 in the radial direction. Concerning the accelerometers, one cause could be inaccuracies in the structural model used to obtain the bending line, as well as in the calibration, which leads to a frequency-dependent error in the low frequency range below the natural frequency. In addition, the placement of the accelerometers, especially the

one measuring tangentially to the tower wall, introduces further uncertainties due to possible misalignment. Concerning the TLS, the uncertainty in the relative position to the turbine as well as the exact measurement height of the TLS should be mentioned. As the laser has a higher signal power than the accelerometers in the range of the first bending natural frequency, it is also possible that the evaluation of the laser data using

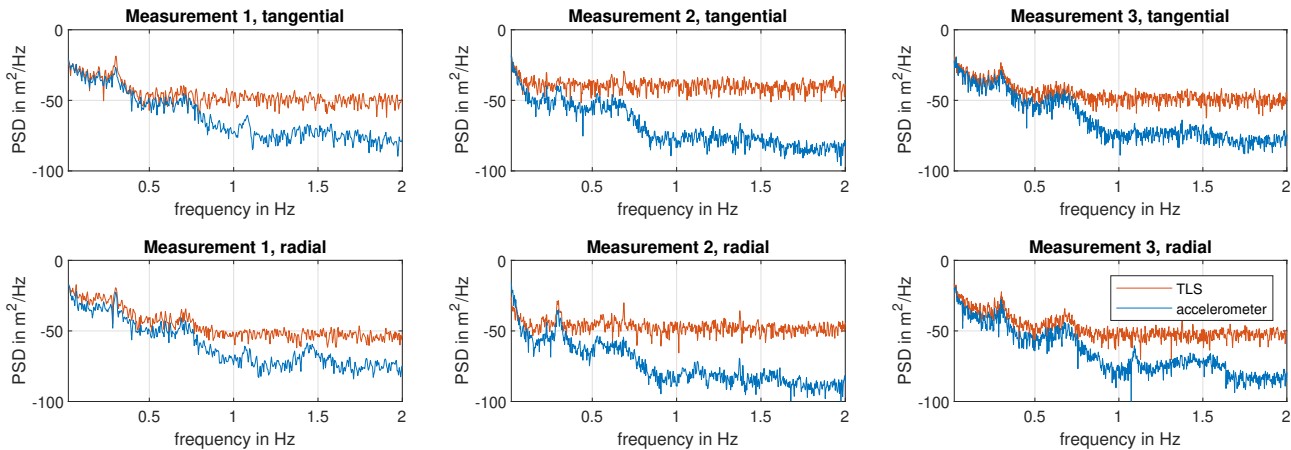

**Figure 16.** Comparison of the power spectral densities (PSD) of the displacements determined with accelerometers and TLS for all measurements in the frequency range from 0.01-2 Hz.

a parabola fit may lead to an overestimation or crosstalk. Despite the uncertainties and deviations mentioned, the tilt error compensation presented appears to function as intended.

## 6 Benefits and Limitations

The results of this study show that using the presented tilt error compensation approach, the different measurement technologies can be successfully aligned. By using a frequency-dependent correction, the accelerometer data is employed for accurate displacement estimations in a significantly expanded low frequency range. Comparisons in the time domain show that residual deviations still remain, which may be attributed to inaccuracies in the positioning of the laser and the accelerometers. Furthermore, inaccuracies in the FE model used to calculate the static bending line and in the algorithm used to extract displacements from the laser measurements contribute to the deviations.

A major benefit of the presented approach is that it can be applied in the monitoring of many slender structures, such as chimneys, offshore wind energy turbine towers or television towers. Similarly to the onshore wind turbine discussed in this paper, these structures exhibit significant structural flexibility and are thus prone to large-amplitude low-frequency motions. The application of the presented approach requires relatively minimal technical equipment, since a two-axis accelerometer measurement system suffices. In addition, only a simple beam model of the structure is required to obtain the static bending line, which is used to parameterise the correction function. Considering the usage of IEPE sensors, a drawback is the high-pass behaviour inherent to the measurement principle, which prevents the estimation of static displacements.

Considering the significant noise level of the contactless TLS-based measurements makes them well-suited for applications where large displacements occur and the measurement noise is not an issue. This is particularly true, e.g., for measuring low frequency displacements, such as those that occur at rotor blades (Helming et al., 2023), where the attachment of a sensor to the measurement object is undesirable. In contrast, accelerometers are preferable for monitoring wind turbine tower dynamics as they have a much lower noise level in the range of the first bending natural frequency and at higher frequencies.

## 7  Summary and Outlook

The tilt error, which leads to the measurement of part of the acceleration due to gravity, has an influence on acceleration measurements in particular in the quasi static motion. Therefore, a tilt error compensation method based on the static bending line and without explicit measuring of the tilt has been introduced. It was shown that the influence of the tilt error on the first normalised bending shape is present with the largest amplitude error of 5% at the lowest measurement level, but it is negligible. A much larger effect is observed in the frequency range below the natural frequency. Below 0.2 Hz the amplitude error is at least 9%. This is particularly pronounced when considering a double time integration to obtain displacements. For quasi-static movements below 0.01 Hz, a 95% overestimation of the dynamic displacement amplitude was observed at the tower head without tilt error compensation, and this error is successfully compensated by using the proposed tilt error compensation approach.

To validate the tilt error compensation, displacement time series determined from accelerometers were compared with displacement time series measured using a TLS in the frequency range from 0.01 Hz to 2 Hz. Except for one measurement in the radial direction of the laser, the basic signal characteristics of both measurement principles are shown to be similar, which indicates that the presented tilt error compensation works appropriately. However, discrepancies still remain, and a precise investigation of the deviations and uncertainties should therefore be carried out in further studies on the basis of previous uncertainty analysis for TLS (Helming et al., 2023) and accelerometers (Jonscher et al., 2022b). In particular, the uncertainty analysis for the tilt error compensation method is still missing and should be carried out by means of in-depth laboratory experiments.

In this study, it has been possible to record the amplitude and phase of low-frequency vibrations down to 0.01 Hz using IEPE accelerometers. To go even lower, DC-capable MEMS accelerometers could be an alternative, despite the higher noise levels. Therefore, it should also be investigated whether the compensation of the tilt error using the method with a bi-axial MEMS accelerometer presented in the study leads to comparable results as using the tilt angle estimated from a tri-axial MEMS accelerometer. Overall, it will be interesting to investigate in future how low-frequency, low-noise MEMS sensors perform in the estimation of dynamic displacements. Since the presented tilt error compensation variant is similar to a second-order high-pass filter

without phase change, this could allow for more low-frequency evaluations than the tilt error compensation based on a direct angle measurement. From a practical point of view, it remains an open question whether a tri-axial MEMS, which is more useful in the approach presented in this study, particularly for compensating for misalignment, or whether the cost savings of a bi-axial MEMS outweigh the disadvantages. Since accelerometers alone can only measure dynamic displacements, sensor fusion techniques with SCADA, or Real-Time Kinematic (RTK) GNSS could be a way to determine very low frequency and quasi-static tower top displacements. This would allow for a better estimation of the loads in the low frequency range without the use of strain gauges.

*Data availability.* The data that support the findings of this study are available from the corresponding author, CJ, upon request.

*Author contributions.* CJ and BH devised the original idea of this research. CJ and PH designed the measurement campaign. CJ, PH and DB were responsible for carrying out the measurement campaign. CJ and PH carrying out the analysis of the measurements. DM and BH created the DeSiO model. AF, TG and RR supervised the work. CJ, PH, DM and AF wrote the manuscript. All authors reviewed the manuscript.

*Competing interests.* RR is associate editor of WES.

## Acknowledgements

We greatly acknowledge the financial support of the Federal Ministry for Economic Affairs and Energy of Germany (research project *PreciWind-Präzises Messsystem zur berührungslosen Erfassung und Analyse des dynamischen Strömungsverhaltens von WEA-Rotorblättern*, FKZ 03EE3013B) and the German Research Foundation (SFB-1463-434502799) that enabled this work. In addition, we are grateful to the *Deutsche Wind-Guard GmbH* for their support during the measurement campaign as well as Max Bögl for information about the hybrid tower.

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
