# Peer review of "Dynamic displacement measurement of a wind turbine tower using accelerometers: tilt error compensation and validation"

_Wind Energy Science, 2023_

## Author Comment (AC1)

**Dynamic displacement measurement of a wind turbine tower using accelerometers: tilt error compensation and validation**

**Answers to the reviewer's comments**

The authors would like to thank the editor and reviewers for their time and effort in reviewing the article and their constructive comments. We appreciate the chance to clarify the points that were commented on. The article is revised according to the reviewers' remarks and queries, with detailed responses and explanations of the edits given below. All edits in the manuscript have been highlighted in red colour.

We note that during the revision, the number of lines has been changed. Therefore, in the following paragraphs, we refer to the revised article.

**Reviewer 1**

**I've enjoyed reading your manuscript on your methods and experimental work on determining the displacement of a turbine tower from IEPE accelerometers and validating your findings using a laser (TLS) setup. The work seems to be conducted in a diligent manner and I have no reason to doubt the conclusions of the work. The authors point out a classic error that is made when using accelerometer data for estimating displacements; the double integration will blow up any low frequency contributions in the spectrum. And in particular because a turbine is slightly tilted (bent) under the nominal load a low frequency contribution from gravity will appear, and exactly this contribution gets blown up by the double integration leading to erroneous estimates of the displacement. In this publication the authors show that it is possible to use a structural model of the turbine to predict this effect on various heights on the structure and introduce a filter to counter this. I like the elegance of this solution and it seems a viable strategy.**

We thank the reviewer for the appreciative review. We hope that our answers to the following points and concerns will clarify why we have submitted the paper to a wind energy-specific journal. The corresponding changes in the manuscript can be found after the answers to comment 3.

**1) However, I do question whether this paper belongs in this journal. The challenges presented are not unique to wind turbines and the paper does not present any results that are unique to wind turbines (E.g. an example on how the method can be used to tackle a particular question in wind energy engineering). Aside for the fact they were collected on a wind turbine no results are linked to the operation or particular dynamics of wind turbines, in fairness it might have been a regular tower.**

We agree with the reviewer that the tilt error is not unique to wind turbine towers but applies to any tower measurement. However, as the reviewer already states, the comparison was carried out on a wind turbine tower. Many readers of the WES journal will find this topic interesting as they might not be fully aware of the significant influence of the tilt error in acceleration measurements on wind turbine structures. Wind turbines are subject to significantly greater head displacements than conventional towers or buildings. The larger these displacements are, the more significant the tilt error becomes. The displacements are caused by the high head mass, the large area exposed to wind in relation to the cross-section of the tower, as well as operational changes to the wind turbine due to, e.g. the yaw movement. These excitations are particularly low-frequency, leading to a significant tilt error, as can be seen in Figure 10. This overestimation leads, for example, to an overly conservative calculation of the remaining service life.

**2) Furthermore the authors motivate the need to resolve this issue to allow for a better estimation of the fatigue life and the displacement. They propose IEPE accelerometers to outperform DC capable sensors, such as MEMS, based on better noise characteristics. While this statement is historically true, high quality MEMS are available today that give IEPE strong competition. And I honestly question whether the differences in signal to noise ratio between a good quality MEMS and IEPE really make any significant difference in the final fatigue/displacement estimate. But if you could show me I'm wrong, then I'm happy to learn.**
**Meanwhile, while the achieved lower frequency bound of the IEPE is impressive, it is still not sufficient to actually capture the slow varying fatigue cycles (e.g. as caused by slow variations in windspeed) that play a significant part in the fatigue of (onshore) wind turbines and would be caught by e.g. MEMS. The authors acknowledge this and mention data-fusion would be required to obtain that goal.**
**Similarly I look at the issue of the tilt error and wonder is this not just an issue of working with just 2 axis? Why could one not look at e.g. Tri-axial MEMS instead? While I didn't do the maths, I suspect that with a Tri-axial MEMS you would be able to compensate for the slow varying tilt error without the need of a model? (but I will accept that I might have overlooked something)**

We concur with the reviewer on the improving noise characteristics of MEMS technology in recent years. Selecting the appropriate sensor technologies and considering the correct frequency ranges are crucial for effective monitoring. Although we employed IEPE sensors due to positive experiences, the reviewer correctly asserts that tilt error measurement is technology-independent. The paper suggests measuring tilt angle as a simple method to address the tilt error, like measuring the vertical acceleration axis using MEMS. However, our paper focuses on a virtual sensing approach for cases where direct tilt angle measurement is not feasible, as often encountered in available wind turbine tower acceleration data, such as Alpha Ventus and WiValdi. We want to process these data records using this method in the future.
The virtual sensor approach enables tilt error compensation using a bi-axial acceleration sensor, regardless of the sensor type. This approach also allows a prediction of the tilt error to determine whether tilt error compensation should be considered in further SHM steps or not. Future research could explore comparing dynamic displacement estimation between bi-axial MEMS sensors and tri-axial MEMS sensors with direct tilt measurement.

**3) To summarize, there is nothing for me to question the actual work done nor the quality thereof or the quality of the measurement. And the paper was clear and easy to read. But as a whole for I didn't feel like it matched this journal and might be more appropriate in e.g. a journal that targets**

**experimental methods. I'm however willing to review a revision of this manuscript if the authors can draw a stronger link with either the final goal of fatigue estimation (e.g. demonstrating an estimate of the quasi-static load from this measurements in comparison to a MEMS based estimate) or a stronger discussion on how wind energy specific phenomena that were initially obfuscated by the tilt error now are made visible.**

In summary, the paper aims to raise awareness that tilt errors influence low-frequency vibrations and that this error can be compensated with a bi-axial accelerometer and knowledge of the bending line. We want to emphasise that the reviewers correctly stated remark that the tilt error compensation is independent of the sensor type and that it was not our intention to compare sensors with regard to the described challenges a tilt error entails. The type of sensor used for a monitoring concept is chosen based on other requirements, such as the frequency range of interest. The paper focuses primarily on dealing with available measurement data that does not include a direct measurement of the tilt angle. With regard to wind energy-specific phenomena, wind turbines are subject to significantly greater head displacements than conventional towers or buildings. Consequently the tilt error plays a significant role for monitoring wind turbines. For a correct dynamic displacement estimation, tilt error compensation is necessary and provides the basis for further methods, such as correctly estimating the remaining useful life. However, the latter is not the objective of this study. We hope we have explained why we chose a wind energy-specific journal to submit this article and why we think readers of the WES journal might be interested in the presented research, especially with the application of the proposed approach on a real wind turbine tower.

To further clarify the aim of the work and the importance of considering the tilt error for wind turbines, we have changed lines 54-78 (pages 2-3) in the manuscript:

*The tilt error is caused by a temporal change in the alignment of the measurement direction relative to the Earth's gravity field, so that a part of the acceleration due to gravity is added to the structural acceleration measured by the sensor (Tarpø et al., 2021), leading to an overestimation. This effect can also be utilised to determine inclination with a tri-axial DC MEMS accelerometer (Luczak, 2014). In the quasi-static case a tower deflects according to the static bending line, so that translatory motion at the tower top is always coupled with a slight tilt of the structure, resulting in a change in orientation relative to the Earth's gravity field. Basically, the tilt error can be corrected using the tilt angle (Boroschek and Legrand, 2006) before integrating the signal in time, such as determining the tilt angle using a tri-axial MEMS sensor. However, the direct measurement of the tilt is not always possible, as vibrations can falsify it. Therefore, Tarpø et al. (2021) developed a geometrical approach to compensate for the tilt error in a laboratory setting. However, if only one bi-axial acceleration sensor per measuring level is provided in the measurement setup, regardless of the sensor type, a virtual sensing concept is required. Since the tilt error is position-dependent on a flexible structure, it can lead to an incorrect identification of mode shapes and overestimation of low-frequency vibration amplitudes. As a consequence of the overestimation of displacement, the structural loads are also overestimated. Although the tilt error occurs in any structure where the orientation of the sensors to the Earth's gravity field changes in the low-frequency range, the tilt error plays a particular role in wind turbines in contrast to other tower structures. This fact is caused by the high head mass, the large area exposed to wind in relation to the cross-section of the tower, as well as operational changes to the wind turbine due to, e.g. the yaw movement. These excitations are particularly low-frequency, leading to a significant tilt error. Therefore, Tarpø et al. (2022) used a virtual sensing approach to estimate the dynamic strain of an offshore wind turbine tower using geophones and removed the tilt error by estimating the tilt angle using tri-axial geophones. So far, the influence of the tilt error on acceleration measurements of wind turbine towers is usually not considered. Furthermore, a predictive compensation method that functions without direct tilt angle measurement and the installation of respective additional sensors yet to be proposed. Such a method is especially relevant for existing measurement set ups, like the first German offshore wind farm Alpha Ventus.*

We have also made it clear in the outlook (lines 355-362, page 19) that, in addition to the approach presented, it is also possible to measure the tipping error directly and that a comparison of both approaches will be useful in the future.

*In addition, the relationship between tower head displacement, wind speed and power should be investigated in future studies. It should also be investigated whether the compensation of the tilt error using the method with a bi-axial MEMS accelerometer presented in the study leads to comparable results as using the tilt angle estimated from a tri-axial MEMS accelerometer. Overall, it will be interesting to investigate in future how low-frequency, low-noise MEMS versions can estimate dynamic displacements. Since the presented tilt error compensation variant is similar to a second-order high-pass filter without phase change, this could allow for more low-frequency evaluations than the tilt error compensation based on a direct angle measurement.*

**Reviewer 2**

**The paper is well-structured, with clear objectives, and it applies valid scientific methods that are well-documented for reproducibility. The results support the authors' interpretations and the discussion is relevant and well-backed. The conclusions are accurately derived from the results.**

We would like to thank the reviewer for the feedback and constructive comments. We have answered them below:
**1)**

- **Technically, the term RUL (Remaining Useful Life) is widespread and shall be used. (See line 16)**

  We have changed the description in the manuscript accordingly (page 1 line 16).

  *For remaining useful life (RUL) estimation, the real loads occurring on the towers are of interest.*

- **Botz (2022) examined the quasi-static tower deflection using RTK. He does not specify standard deviation and quality flag of the solution. In general, RTK has long-term stable static precision. Care must be taken what data is precise enough. My personal experience from a case study is, that less than 20% of the measured data has a standard deviation ¡ 0.02 m and a quality flag of 1 (fixed solution)**

  We thank the reviewer for sharing his valuable experience. We have added the following to the manuscript (page 21 line 24-26):

  *Except for GNSS, which in the case of Botz (2022) indicates unrealistically high quasi-static deformations for the top of the tower without specifying the quality flag or standard deviation, all other approaches require a reference point.*

- **Line 39: "by using of" revise for clarity**

  We have deleted the "of".

- **Line 116: "The time integration of measurement data can also be performed in the time domain" revise for clarity**

  We have changed in the manuscript (page 5 lines 124):

  *The time integration of the acceleration data can also be carried out in the time domain by using filters.*

- **Figure 3: axis description missing**

  We have labelled the axes and changed the caption as follows (page 8):

[Figure]

Figure 1: Double logarithmic representation of the frequency-depended correction factor of $a_{\mathrm{meas}}$ as a function of frequency for $gm = 0.4267 \; \frac{1}{s^2}$. There is no phase change due to the tilt error compensation.

**2) Tilt error is indeed of great importance for dynamic displacement estimation in time domain. I strongly agree with the importance of measures for tilt error. As the tilt error strongly depends on the nacelle and rotor weight induced bending but also the thrust force induced bending, I'd love to see the SCADA data (wind and power output as a comparison to the plots. It would be important, especially for Figure 13.**

Unfortunately, we only have aggregated 10-minute SCADA. Therefore, we only have one value for each measurement, which are listed in Table 2. However, the relationship should be investigated in future studies. We added the power in Table 2 and added in the caption of Figure 14 in the manuscript (page 16):

*Comparison of the displacements determined from accelerometers (frequency range 0.005-2 Hz) and TLS (frequency range 0-2 Hz) for TLS measurement 1 with a averaged wind speed of 12.53 m/s and power of 3373 kW.*

We have also added in the manuscript (page 11 lines: 231-232):

*Only this 10 minutes aggregated SCADA data are available during the measurement campaign.*

We have also added in the manuscript (page 19 lines: 355-356):

*In addition, the relationship between tower head displacement, wind speed and power should be investigated in future studies.*

**3) I would recommend a flowchart for the data pipeline, as this is hard to understand from text only.**

We thanks the reviewer for the suggestion and agree with the comment. We have added the Figure 2 (page 4) and the following to the manuscript (page 4 lines 92-93):

[Figure]

Figure 2: *Flowchart of the dynamic displacement determination from acceleration measurement data taking into account the tilt error compensation based on the static bending line.*

*This study describes how low-frequency displacements of towers can be estimated using accelerometers, taking into account the tilt error. The corresponding flowchart of the process is shown in Fig. 2. The required theory of the various process steps is described in terms of accelerometers, integration filters and the compensation of the tilt error based on the static bending line.*

---

## Referee Report (RR1)

**1) Scientific significance:**
Does the manuscript represent a substantial
contribution to scientific progress within the scope
of Wind Energy Science (substantial new concepts,
ideas, methods, analyses, or data)?

○ Excellent ● Good ○ Fair ○ Poor

**2) Scientific quality:**
Are the scientific approach and applied methods valid?
Is sufficient information given so other
researchers (in principle) can repeat the work? Are the
results discussed in an appropriate and
balanced way (consideration of related work, including
appropriate references)?

○ Excellent ● Good ○ Fair ○ Poor

**3) Presentation quality:**
Are the scientific results and conclusions presented in
a clear, concise, and well-structured way
(abstract conveys efficiently the essence of the paper;
number and quality of figures/tables;
appropriate, fluent and precise use of English
language)?

○ Excellent ○ Good ○ Fair ● Poor

**For final publication, the manuscript should be**

○ **accepted as is**.

○ accepted subject to **technical corrections**.

○ accepted subject to **minor revisions**.

● reconsidered after **major revisions**:

○ **rejected**.

**Were a revised manuscript to be sent for another round of reviews:**

● I would be willing to review the revised manuscript.

○ I would not be willing to review the revised manuscript.

The paper

"Dynamic displacement measurement of a wind turbine tower using accelerometers: tilt error compensation and validation"

By

Clemens Jonscher et al.

focuses on vibration-based structural health monitoring (SHM) for wind turbine support structures, specifically addressing the challenge of tilt error in accelerometer measurements. Tilt error arises from the gravitational acceleration component mixed with structural acceleration data. If not accounted for, this error can lead to inaccurate evaluations, especially in fatigue estimation and dynamic displacement assessments.

Traditional methods require an additional sensor to measure the tilt angle at each point, which is not feasible for previously recorded measurements lacking tilt information. To address this, the study proposes a novel tilt error compensation method using the static bending line of the structure. This approach allows for estimating the tilt error in advance without the need for extra sensors.

The method was validated on an onshore wind turbine tower using accelerometer measurements and compared with contactless absolute distance measurements obtained via terrestrial laser scanning (TLS). Results show that the tilt error significantly impacts quasi-static motion below 0.2 Hz, leading to a minimum amplitude error of 9%. However, normalized bending mode shapes around 0.3 Hz are only slightly affected.

In summary, the proposed tilt error compensation method is interesting and could be vey useful to researchers and practitioners alike. However, before granting full acceptance, the following remarks should all be addressed by the Authors.

1. A flowchart of the complete method, building upon the one already reported in Figure 1 for signal preprocessing, could improve the reader's understanding, , reporting in a clearer and visual way a step-by-step breakdown of the methodology
2. The authors could provide more context for the amplitude error percentages (e.g., 9% and 95% errors) and how these values impact the practical application of the method. Do these errors have significant consequences in real-world monitoring for daily operations?
3. If this Reviewer understood well the Authors' intended meaning, the "95% overestimation of dynamic displacement amplitude" at low frequencies is quite high. Addressing whether this error can be reduced, or how it might impact the overall results, would strengthen the analysis.
4. It would be useful to further expand on how noise is currently handled in the TLS data and accelerometers, and what further preprocessing techniques can be applied to minimize it.
5. Further elaboration on how this tilt compensation method could be integrated with existing structural health monitoring (SHM) systems (e.g., in wind turbines) would be useful. Discussing how feasible the implementation is, whether it requires specific hardware, and any practical challenges would add practical value. In this regard, it can be useful to introduce and mention the recent review works of https://doi.org/10.3390/s22041627, where several technologies and strategies are discussed.
6. If this Reviewer understood correctly, the idea of using bi-axial versus tri-axial MEMS accelerometers for tilt error compensation is interesting but underexplored. More details on this comparison and how it might influence future designs would enhance the discussion.
7. It would be beneficial to include error bars or confidence intervals for the measurements, providing a more robust quantification of uncertainty. Also visualizing these C.I.s would be useful.
8. The Conclusions are a bit lengthy and could be shortened.
9. The conclusions touch on some potential limitations (e.g., inaccuracies in the laser and accelerometer positioning, deviations from the FE model). These could be highlighted more explicitly as limitations in the paper, so readers understand the scope and limitations of the proposed method.
10. The conclusions suggest that the method works well for wind turbine tower dynamics but could be expanded to other applications (e.g., rotor blades or other low-frequency displacement measurements). Providing a clearer discussion on the method's scalability to other structures or conditions would add value to the paper.

11. The applicability of the method for different frequencies and measurement environments (such as offshore wind turbines) could also be discussed in more detail.

---

## Author Response (AR2)

**Dynamic displacement measurement of a wind turbine tower using accelerometers: tilt error compensation and validation**

**Answers to the reviewer's comments**

The authors would like to thank the editor and reviewer for their time and effort in reviewing the article and their constructive comments. We appreciate the chance to clarify the points that were commented on. The article is revised according to the reviewers' remarks and queries, with detailed responses and explanations of the edits given below. All edits in the manuscript have been highlighted in red colour.

We note that during the revision, the number of lines has been changed. Therefore, in the following paragraphs, we refer to the revised article.

**Reviewer**

**In summary, the proposed tilt error compensation method is interesting and could be very useful to researchers and practitioners alike. However, before granting full acceptance, the following remarks should all be addressed by the Authors**

We appreciate your helpful comments. We hope that the revision will significantly improve the understanding of the manuscript.

**1) A flowchart of the complete method, building upon the one already reported in Figure 1 for signal preprocessing, could improve the reader's understanding, reporting in a clearer and visual way a step-by-step breakdown of the methodology**

We have updated Figure 1 and, in addition to the dynamic displacement determination process, we have also included the laser validation measurement and added visualisations.

**2) The authors could provide more context for the amplitude error percentages (e.g., 9% and 95% errors) and how these values impact the practical application of the method. Do these errors have significant consequences in real-world monitoring for daily operations?**

Indeed, these amplitude errors significantly impact fatigue life estimation, if they are not accounted for. Therefore, a numerical example using the quoted figures (9% and 95%), which are also shown in Figure 10, was added to the paper to exemplify this relation.

*For very low frequencies, Fig. 10 demonstrates, that the correction factor of the highest measuring level drops below 0.05. This means that 95% of the measured acceleration amplitude is due to tilt and only 5% are due to the actual structural acceleration. The impact of these errors can be exemplified for tower fatigue life estimation as a typical application for structural measurements. The fatigue life ratio between the corrected and raw measurement data can be expressed, assuming small deformations, a linear stress-strain relationship and a harmonic acceleration signal, using an equation inspired by the Eurocode 3 (EN 1993-1-9:2005).*

$$\frac{N_{corr}}{N_{raw}} = \left(\frac{\Delta\sigma_{raw}}{\Delta\sigma_{corr}}\right)^m = \left(\frac{a_{raw}}{a_{corr}}\right)^m = (c(f))^m, \tag{1}$$

*where $m = 3$ for steel structures and $\frac{a_{raw}}{a_{corr}}$ corresponds to the correction factor $c(f)$. Under the assumptions outlined above, the ratio between the measured and corrected stresses $\frac{\Delta\sigma_{raw}}{\Delta\sigma_{corr}}$ also corresponds to $c(f)$. Using the correction factor $c(f) = 0.91$ results in $\frac{N_{corr}}{N_{raw}} = 0.75$, which means that using the uncorrected data results in 25% reduced fatigue life estimates. Considering $c(f) = 0.05$ results in $\frac{N_{corr}}{N_{raw}} = 0.00013$, which shows that for very low frequencies, the raw data is essentially unusable for fatigue life estimations.*

**3) If this Reviewer understood well the Authors' intended meaning, the "95% overestimation of dynamic displacement amplitude" at low frequencies is quite high. Addressing whether this error can be reduced, or how it might impact the overall results, would strengthen the analysis.**

The stated overestimation of the dynamic displacement amplitude is fully compensated by the proposed tilt error compensation approach. The wording was changed to make this clearer.

*For quasi-static movements below 0.01 Hz, a 95% overestimation of the dynamic displacement amplitude was observed at the tower head without tilt error compensation, and this error is successfully compensated by using the proposed tilt error compensation approach.*

**4) It would be useful to further expand on how noise is currently handled in the TLS data and accelerometers, and what further preprocessing techniques can be applied to minimize it.**

The best way to reduce noise in the measurement is to use measurement technology with as little noise as possible. Noise can be reduced after the measurement by digital filtering by filtering out frequency components that are not relevant to the measurement task. In this study, only frequency components up to 2 Hz were considered for both measurement methods to avoid high-frequency noise. To clarify this in the manuscript, the following has been added:

*The bandpass filter is used to ensure that only frequency components relevant for the evaluation are included in the time signal by reducing measurement noise.*

**5) Further elaboration on how this tilt compensation method could be integrated with existing structural health monitoring (SHM) systems (e.g., in wind turbines) would be useful. Discussing how feasible the implementation is, whether it requires specific hardware, and any practical challenges would add practical value. In this regard, it can be useful to introduce and mention the recent review works of https://doi.org/10.3390/s22041627, where several technologies and strategies are discussed**

Thank you for these valuable hints. The method requires bixaxial acceleration sensors that measure horizontally. These are usually already installed in the nacelle, as the VDI 3834 recommends the use of acceleration sensors for monitoring wind turbines. We have added the following to the introduction in the manuscript:

*There are other monitoring approaches using different sensor technology as described by Civera and Surace (2022). However, the VDI 3834 recommends using the nacelle acceleration to estimate the health status using vibration measurements. Accelerometers are therefore already installed in many wind turbines.*

**6) If this Reviewer understood correctly, the idea of using bi-axial versus tri-axial MEMS accelerometers for tilt error compensation is interesting but underexplored. More details on this comparison and how it might influence future designs would enhance the discussion.**

The presented tilt error compensation approach requires only measurement data in the horizontal directions. Three-axis sensors are therefore not required in principle. However, for practical applications, it is still beneficial to install three-axis sensors, since the additional measurement axis provides valuable data on sensor misalignment and also enables plausibility checks due to redundancy. We have added the following to the summary and outlook Section:

From a practical point of view, it remains an open question whether a tri-axial MEMS, which is more useful in the approach presented in this study, particularly for compensating for misalignment, or whether the cost savings of a bi-axial MEMS outweigh the disadvantages.

**7) It would be beneficial to include error bars or confidence intervals for the measurements, providing a more robust quantification of uncertainty. Also visualizing these C.I.s would be useful.**

Thank you for this hint. We decided to not show confidence intervals or error bars in the time series analysis, as the uncertainty intervals of the amplitude and phase shift are frequency-dependent. However, for the TLS measurement and acceleration sensor calibration, uncertainty analyses for the measurement methods were carried out in preliminary work (Helming et al., 2023 and Jonscher et al., 2022b). And to get a better impression of the effectiveness of the method and the similarity of the signals, we have added the Table 4 and the following paragraph:

*A comparison based on the Pearson correlation coefficient and the normalised mean square error (nMSE), given by:*

$$nMSE = \frac{1}{N\sigma^2_{w_{laser}}} = \sum_{i=1}^{N} \left( w_{laser,i} - w_{accel,i} \right)^2, \tag{2}$$

*is listed for all three measurements in Table 4 for the case with and without the presented tilt error compensation of the acceleration measurement. A Pearson correlation coefficient closer to 1 indicates that the signals are correlated, with a value close to 0 the signals are uncorrelated. Tilt error compensation makes the signals - with the exception of measurement two in radial direction - significant more correlated. However, this metric cannot provide any information about an amplitude error. The nMSE is used for this purpose. It is clearly evident that the error is greatly reduced by the proposed tilt error compensation method.*

In the outlook, we have also made the missing uncertainty analysis even clearer:

*However, discrepancies still remain, and a precise investigation of the deviations and uncertainties should therefore be carried out in further studies* *on the basis of previous uncertainty analysis for TLS (Helming et al., 2023) and accelerometers (Jonscher et al., 2022b). In particular, the uncertainty analysis for the tilt error compensation method is still missing and should be carried out by means of in-depth laboratory experiments.*

**8) The Conclusions are a bit lengthy and could be shortened.**

The conclusion was shortened and some content was moved to the newly created section on benefits and limitations.

**9) The conclusions touch on some potential limitations (e.g., inaccuracies in the laser and accelerometer positioning, deviations from the FE model). These could be highlighted more explicitly as limitations in the paper, so readers understand the scope and limitations of the proposed method.**

The remarks on the limitations of the measurement setups and modelling were moved to the newly created section on benefits and limitations:

*The results of this study show that using the presented tilt error compensation approach, the different measurement technologies can be successfully aligned. By using a frequency-dependent correction, the accelerometer data is employed for accurate displacement estimations in a significantly expanded low frequency range. Comparisons in the time domain show that residual deviations still remain, which may be attributed to inaccuracies in the positioning of the laser and the accelerometers. Furthermore, inaccuracies in the FE model used to calculate the static bending line and in the algorithm used to extract displacements from the laser measurements contribute to the deviations.*
*A major benefit of the presented approach is that it can be applied in the monitoring of many slender structures, such as chimneys, offshore wind energy turbine towers or television towers. Similarly to the onshore wind turbine discussed in this paper, these structures exhibit significant structural flexibility and are thus prone to large-amplitude low-frequency motions. The application of the presented approach requires relatively minimal technical equipment, since a two-axis accelerometer measurement system suffices. In addition, only a simple beam model of the structure is required to obtain the static bending line, which is used to parameterise the correction function. Considering the usage of IEPE sensors, a drawback is the high-pass behaviour inherent to the measurement principle, which prevents the estimation of static displacements.*
*Considering the significant noise level of the contactless TLS-based measurements makes them well-suited for applications where large displacements occur and the measurement noise is not an issue. This is particularly true, e.g., for measuring low frequency displacements, such as those that occur at rotor blades (Helming et al., 2023), where the attachment of a sensor to the measurement object is undesirable. In contrast, accelerometers are preferable for monitoring wind turbine tower dynamics as they have a much lower noise level in the range of the first bending natural frequency and at higher frequencies.*

**10) The conclusions suggest that the method works well for wind turbine tower dynamics but could be expanded to other applications (e.g., rotor blades or other low-frequency displacement measurements). Providing a clearer discussion on the method's scalability to other structures or conditions would add value to the paper.**

A remark on the transferability to different structures was added to the newly created section on benefits and limitations. See in the changes of remark number 9.

**11) The applicability of the method for different frequencies and measurement environments (such as offshore wind turbines) could also be discussed in more detail.**

As the eigenfrequencies of onshore and offshore turbines only differ around a factor of two, there are no major obstacles to the application of the proposed approach to offshore turbines. Wave spectra rarely possess significant components below $0.1\,\mathrm{Hz}$, hence tower movements due to wave interaction are expected to be captured well. Offshore wind turbines were therefore added to the revised section references in remark number 9.